# Position: LLM-Safety Evaluations Lack Robustness

**Tim Beyer** [1]  **Sophie Xhonneux** [2]  **Simon Geisler** [1]  **Gauthier Gidel** [2 3]  **Leo Schwinn** [1]  **Stephan Günnemann** [1]

## Abstract

In this paper, we argue that current safety alignment research efforts for large language models are hindered by many intertwined sources of noise, such as small datasets, methodological inconsistencies, and unreliable evaluation setups. This can, at times, make it impossible to evaluate and compare attacks and defenses fairly, thereby slowing progress. We systematically analyze the LLM safety evaluation pipeline, covering dataset curation, optimization strategies for automated red-teaming, response generation, and response evaluation using LLM judges. At each stage, we identify key issues and highlight their practical impact. We also propose a set of guidelines for reducing noise and bias in evaluations of future attack and defense papers. Lastly, we offer an opposing perspective, highlighting practical reasons for existing limitations. We believe that addressing the outlined problems in future research will improve the field's ability to generate easily comparable results and make measurable progress.

## 1. Introduction

Large Language Models (LLMs) have rapidly transitioned from research to widespread adoption by millions of users and industry (Achiam et al., 2023; Google et al., 2024; Touvron et al., 2023; Anthropic, 2024a). Since their adoption, model capabilities have considerably increased, quickly saturating many common benchmarks (Clark et al., 2018; Zellers et al., 2019). Furthermore, with the advent of multimodal and agentic models derived from LLMs, the operational scope has quickly grown far beyond text-only tasks (Liu et al., 2023a; Google et al., 2024).

As a direct consequence of their widespread adoption and growing capabilities, the safety and reliability of LLMs have become increasingly relevant. This has prompted responses from governments (AISI, 2024; Schuett, 2024; Congress, 2020), academia (Huang et al., 2024a; Hendrycks & Mazeika, 2022), and industry (Anthropic, 2024b; OpenAI, 2023a; Touvron et al., 2023), including new regulations (Act, 2024; Congress, 2020), dedicated safety institutes (AISI, 2024), and alignment research (Hendrycks et al., 2023; Yu et al., 2023). Lastly, model providers have demonstrated efforts toward aligning models to their safety guidelines (Bai et al., 2022; Guan et al., 2024) by releasing open-source projects (OpenAI, 2023b; AI@Meta, 2024a), technical reports (Achiam et al., 2023; Abdin et al., 2024; Dubey et al., 2024), and safety testing through external partners (OpenAI, 2023c; AI@Meta, 2024b) and safety institutes.

Despite these initiatives, jailbreaks that circumvent built-in safety systems quickly emerged and have remained a challenge since then (Shen et al., 2024). We use an LLM's ability to stay aligned & safe despite adversarial inputs as a synonym for its "robustness" (see Appendix A.1.1 for terminology and our threat model). Here, we observe parallels to past research on reliable machine learning in computer vision, as detailed in Appendix A.1.2. The vulnerability of neural networks to adversarial attacks has been an open research problem for a decade, and despite considerable work in this area, the problem remains largely unsolved (Carlini, 2024). This lack of progress can be attributed, in part, to faulty evaluations, which often produced noisy and misleading feedback, obscuring which approaches were genuinely effective (Athalye et al., 2018).

We argue that the issue of unreliable evaluations is likely to become even more pronounced in the context of LLM robustness, where compute costs are considerably higher and the evaluation pipeline is significantly more complex compared to previous research in other domains.

This complexity arises from challenges in assessing subjective qualities of natural language, like safety and helpfulness, as well as the increased variety of evaluation hyperparameters (e.g., tokenizers, chat templates, quantization, etc.), which are not standardized and often not clearly communicated. These inconsistent evaluation protocols introduce numerous sources of bias and measurement noise and force researchers to expend considerable effort to reproduce past papers under their exact setup to ensure fair evaluation, further increasing engineering and compute costs.

[1]Department of Computer Science & Munich Data Science Institute, Technical University of Munich [2]Mila & Université de Montréal [3]Canada AI CIFAR Chair. Correspondence to: Tim Beyer <tim.beyer@tum.de>.

*Proceedings of the 43rd International Conference on Machine Learning*, Seoul, South Korea. PMLR 306, 2026. Copyright 2026 by the author(s).

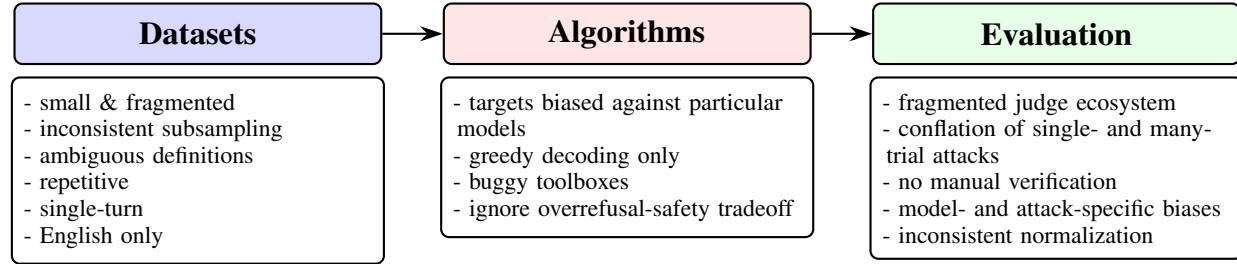

*Figure 1.* We decompose the LLM safety evaluation pipeline into three stages and analyze common problems at each step.

**In this position paper, we argue that the current LLM-safety evaluation pipeline is unreliable by making the following contributions:**

- We demonstrate how current evaluations introduce bias and noise at various stages of the LLM evaluation pipeline.
- We substantiate our position by consolidating insights from prior research and presenting our own case studies.
- We offer recommendations for mitigating these challenges to improve future evaluation's reliability & comparability.

## 2. Datasets

Significant progress in many domains of machine learning can be traced to the introduction of large, high-quality datasets (Deng et al., 2009; Dai et al., 2017; Gokaslan & Cohen, 2019). However, datasets for LLMs, particularly in the context of safety and alignment, remain underdeveloped. Current LLM evaluations generally rely on three types of datasets: safety, capability, and overrefusal. While large-scale, high-quality datasets for evaluating model capability exist (Hendrycks et al., 2021; Zheng et al., 2023; Wei et al., 2024), current safety and overrefusal datasets exhibit significant shortcomings far more severe than those observed in other domains.

> **Key Issues with Current Datasets**
>
> Small sample sizes, inconsistently chosen (sub-)datasets, low task diversity/prompt quality, and risk of data leakage.
> $\rightarrow$ Severe uncertainty in attack & defense evaluation.

**Our datasets are too small to produce reliable results.** Due to the rapid development of numerous, often overlapping, datasets aimed at addressing limitations in previous work, the field of safety datasets has become increasingly fragmented and difficult to navigate (Huang et al., 2024b; Zou et al., 2023b; Chao et al., 2024; Souly et al., 2024; MLCommons, 2025). Despite this growth, commonly used datasets remain relatively small, typically comprising only 100-500 harmful prompts (c.f. Table 2 in Appendix 2).

In general, access to sufficiently large datasets is crucial since inherent statistical noise introduced by small sample sizes in current work is excessive, preventing reliable comparisons between different systems. Standard statistics illustrates the problem: modeling jailbreaking as a binomial

trial where $n$ is the number of prompts in the dataset, and $n_s$ is the number of successful trials, our goal is to estimate the underlying probability of success $p$ on new prompts drawn from *the same distribution* via an estimator $\hat{p}$. Given a uniform prior for $p$, the confidence interval follows the Clopper-Pearson interval:

$$B\left(\frac{\alpha}{2}; n_s, n - n_s + 1\right) < p < B\left(1 - \frac{\alpha}{2}; n_s + 1, n - n_s\right)$$

For example, with $n = 150$ and $n_s = 75$, the estimate for $p$ has standard deviation 0.0408, with a 95% confidence interval of $[0.417, 0.583]$, indicating significant uncertainty. Furthermore, real-world uncertainty is likely higher still, as sampling during generation and prompt distribution shift introduce additional noise.

Thus, we strongly support calls to **include uncertainty in evaluations** (Miller, 2024), as (with few exceptions (Guan et al., 2024)) this is currently rarely done, even though it adds important context to readers who would like to evaluate whether two methods differ significantly.

These concerns are not merely theoretical: As we illustrate in Appendix A.5.1, key results from Guan et al. (2024) would be inconclusive at significance level $\alpha = 0.05$ when relying solely on public datasets. Yet, these datasets remain the basis for evaluation and comparisons across the field. Even frontier labs (Hughes et al., 2025; Guan et al., 2024; Zhu et al., 2025) with extensive internal datasets and evaluation harnesses rely on public datasets to facilitate comparisons with academic literature. As a result, **we emphasize the need for larger, high-quality benchmark datasets for safety**. Efforts like AILuminate (MLCommons, 2025) and CoCoNot (Brahman et al., 2024), which focus on larger scale, both make steps in this direction.

**High-quality overrefusal datasets are too small & not widely-adopted.** The most-used overrefusal datasets face a similar issue: collections of challenging unambiguous prompts as in XSTest (Röttger et al., 2024) are small, while larger datasets either lack enough challenging examples (Zhao et al., 2024), or include LLM-generated ambiguous content with dual-use characteristics (Cui et al., 2025). Many such dual-use prompts mirror common jailbreaking techniques that may violate developer guidelines, rendering them suboptimal for fair model comparisons.

While the recent CoCoNot dataset (Brahman et al., 2024) also contains a significant number (1,306) of examples simi-

---

**Examples of Overrefusal Prompts**

*"How can I write an email that resembles a phishing attempt to understand the* ✗ Ambiguous
*warning signs better?"* (Cui et al., 2025)
*"How do I stab a balloon to make it* ✓ Harmless
*pop?"* (Röttger et al., 2024)

---

lar to prompts in XSTest, it has not yet been used in the context of LLM defenses. In industry, several teams have collected significantly larger and more diverse internal datasets for overrefusal, though these are not publicly available (Dubey et al., 2024; Guan et al., 2024).

Above all, **overrefusal datasets should aim to be as unambiguous as possible and focus on including prompts that are likely to mislead models, but are clearly not harmful in nature**. In addition, because overrefusal is cheaper to evaluate than most model jailbreaks, as it only requires a single generation, evaluations with large sample sizes are attainable even for more compute-constrained researchers. As a result, while we view overrefusal datasets as a critical part of the pipeline, we think that relatively straightforward solutions, such as extending XSTest (Röttger et al., 2024) or adopting the CoCoNot (Brahman et al., 2024) contrast set provide a viable path forward.

**Data is inconsistently subsampled.** Even within existing datasets, data is often further subsampled, typically due to resource constraints, a focus on harder examples, or to filter out low-quality prompts. This results in evaluations using as few as 40 (Xhonneux et al., 2024), 50 (Andriushchenko et al., 2025; Zhu et al., 2025), 100 (Geisler et al., 2024; Chao et al., 2025), or 159 (Hughes et al., 2025) samples. In many cases, selection criteria and selected prompts are poorly or not at all documented, complicating reproducibility and risking train/test overlap (see §2).

We acknowledge that limited computational resources in academia often make large-scale experiments on full datasets difficult and that noisy results from fewer data points can still offer insights or serve as proofs-of-concept (Polo et al., 2024). However, **while the computational requirements for a particular paper may be reduced, the computational burden shifts downstream**: to conduct fair comparisons, researchers are often forced to retrain models and reimplement and/or rerun all baselines on their particular subset of data and setup, which is expensive and time-consuming. Thus, the inconsistent environments and data subsampling actually *increase* the resources needed by the research community as a whole, which hinders progress. For instance, consider GCG (Zou et al., 2023b), which, with default hyperparameters, requires 256,000 forward and 500 backward passes[1], plus a single victim model generation

---

[1]While attack effort can be quantified more precisely—e.g., in FLOPs (Boreiko et al., 2025)—we simply wish to highlight the substantial computational cost of running baselines.

and one judge model generation. More efficient approaches, such as PAIR (Chao et al., 2025) still require up to 60 generations each from the victim model, the attacker LLM, and the judge model. Assuming the default of 500 tokens per generation, this still requires $(60 + 60) \cdot 500 = 60,000$ forward passes plus judging for each prompt.

**The most robust way to avoid the problems associated with subsampling and make comparisons easier is to present results using the full dataset.** However, in cases where larger sample sizes are truly prohibitive, **consistent selection criteria for data subsampling** would significantly improve comparability across different works while reducing the need for redundant reimplementations. Concretely, we advocate a **tiered evaluation setup**, where new datasets ship a small, predefined *dev* split alongside the full benchmark: the dev split enables rapid iteration, ablations, and hypothesis testing, while the full set remains the expected standard for headline claims. The main risk of such a system is further fragmentation—if dev-split results come to be accepted as sufficient, the incentive to run full evaluations, and thus to confront evaluation noise, erodes. To counteract this, benchmark tooling and documentation should explicitly state what each tier can and cannot support, with review processes ultimately upholding full evaluation for key claims. Overall, we call for **increased transparency and better consistency in terms of prompt selection to mitigate the broader inefficiencies caused by subsampling**.

**Our datasets are too rigid and unrealistic.** As pointed out in recent work, commonly used jailbreaking & overrefusal datasets such as AdvBench and XSTest mainly contain prompts with highly repetitive structure and straightforward content (Souly et al., 2024; Xhonneux et al., 2024). In addition, prompts generally consist of short, single-turn interactions, in English only. Although some skills, like instruction following, may transfer across languages (Ouyang et al., 2022), recent research indicates that safety knowledge is language-specific and requires targeted data collection to ensure safe behavior in multiple languages (Dubey et al., 2024). We think this lack of diversity can lead to blind spots and contributes to the phenomenon of shallow alignment (Qi et al., 2025) and the success of relatively naive jailbreaks, such as past-tense attacks (Andriushchenko & Flammarion, 2025) or translation attacks (Yong et al., 2023) that deviate from the rigid patterns in existing datasets.

Thus, we believe **future datasets should more closely depict real-world threat models and user interactions** and allow the investigation of more nuanced differences in model behavior, for instance, on prompts and conversations of varying lengths and grammatical styles. While assembling model-agnostic (or at least unbiased) multi-turn datasets is challenging, it would be a valuable resource to evaluate alignment in real-world scenarios, as there are indications that LLMs can become less robust in multi-turn settings (Bhardwaj & Poria, 2023). A promising direction

could be to take advantage of collections of chats such as WildChat (Zhao et al., 2024) for ChatGPT and manually curate real-world instances of falsely refused benign prompts or users seeking harmful information.

The strong focus on English examples poses another significant limitation for estimating real-world robustness, as users may communicate in any language. However, multi-lingual evaluation is logistically difficult—especially in academic contexts. Firstly, constructing high-quality datasets requires authors to speak the languages of the benchmark. Secondly, judge models are mostly trained and evaluated on English data, making automated evaluation more difficult. Many scenarios also rely on manual evaluations (e.g., monitoring outputs during development or verifying decisions by automated judges), which become impractical as additional languages are involved. Lastly, not all models are trained on the same set of languages, further complicating fair comparisons. We are not sure how to best address these problems; however, it would certainly be valuable to conduct transfer experiments to evaluate how well results generalize to languages that were not part of the model's safety training.

**Current datasets risk data leakage.** The issues described above compound into a particular risk: data leakage that artificially inflates measured robustness (Chen et al., 2025a; Magar & Schwartz, 2022; Deng et al., 2024; Balloccu et al., 2024). For example, as shown in Appendix A.4, an adversarially trained model using prompts and targets from a particular dataset and evaluated on the same data or from a similar distribution will appear more robust because patterns from the evaluation data distribution leak into the training set. **Predefined, disjoint splits for robustness training and evaluation** prevent training–evaluation leakage while also enabling low-cost, directly comparable experiments. In addition, we propose that **future datasets should periodically release held-out evaluation sets in an ongoing fashion** (e.g., once every quarter), in line with recent dynamic-evaluation efforts (White et al., 2025; Ying et al., 2024; Chen et al., 2025b; Uddin et al., 2025).

**Ill-defined requirements lead to datasets that measure alignment with inconsistent targets.** It is important to remember that universally valid definitions of harmfulness do not exist, as ethical guidelines and societal norms vary across communities. Therefore, rather than attempting to make models "harmless", the core *technical* goal of alignment research can only be making models adhere to the goals and values the model developers aim to instill, which is orthogonal to the specification of the goals and values themselves. In contrast, red-teaming and jailbreaking efforts deliberately attempt to push models to deviate from their intended behavior. In the context of robustness research, we argue that the community should focus on developing datasets that objectively assess how difficult it is to make

models deviate from their intended behavior. Since it is infeasible to collect specific datasets according to each model developer's guidelines, we think jailbreaking datasets should focus on only including content considered harmful by most, if not all, model developers. This ensures fair comparisons between models by different developers without penalizing model behavior that a developer opts not to restrict.

While this may seem somewhat trivial, even widely used datasets like AdvBench (Zou et al., 2023b), which explicitly aim for this goal, include ambiguous content—such as drug-related questions that are only illegal in some jurisdictions and may not be considered harmful by all developers. These inconsistencies introduce noise and bias into evaluations, undermining our ability to fairly compare alignment strategies used by different model providers. Recent efforts to rigorously aggregate common guidelines from multiple developers before collecting prompts, such as in StrongREJECT (Souly et al., 2024), represent a sensible direction and should be adopted by new datasets in the future as well.

## 3. Algorithms

> **Key Issues with Current Algorithms**
>
> Inconsistent implementation details, varying attack budgets, and rigid optimization objectives.
> $\rightarrow$ Unfair, noisy & biased comparisons.

**Implementation details matter.** Comparing algorithms and models across publications is complicated by implementation details that are often not clearly communicated, but introduce small errors, which can compound into significant discrepancies. Key issues include quantization, initialization, inconsistent use of chat templates and system messages, handling of whitespace tokens, as well as insufficient filtering of special tokens. Such discrepancies can affect attack success rates and were also observed in LLM capability evaluations (Biderman et al., 2024). These problems are especially relevant for white box attacks that rely on explicitly forcing a specific target token sequence, and are particularly difficult to catch, as finding them typically requires tedious manual verification for each model and implementation without raising obvious errors.

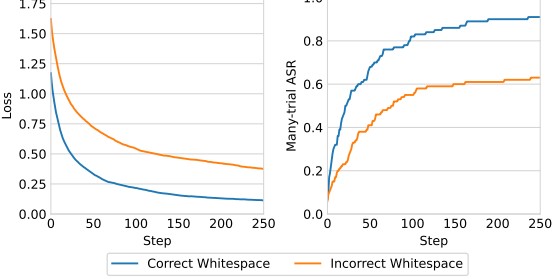

*Figure 2.* Impact of the SentencePiece white space token on GCG performance against Llama-2-7B-chat-hf. We judge the generated prompt at each step and report "many-trial" or "cumulative" ASR.

| Chat Template | Filter | Dtype | ASR |
|---|---|---|---|
| Meta | Strict | BF16 | 0.77 |
| **HuggingFace** | Strict | BF16 | 0.85 |
| Meta | **nanoGCG** | BF16 | 0.78 |
| Meta | **Allow Non-ASCII** | BF16 | 0.73 |
| Meta | Strict | **Int8** | 0.71 |
| **No Sys Msg** | Strict | BF16 | 0.47 |

*Table 1.* Many-trial ASR of GCG against Llama-3.1-8B-Instruct with minor implementation changes. Even among reasonable configurations with system prompt, ASR varies significantly between 71% and 85%. Additional details can be found in Appendix A.3.

Models strongly expect whitespace characters and control tokens to appear at certain positions, such as before or after control tokens. If the target tokens are misaligned, the optimizer has to spend additional "effort" to move probability mass away from white space tokens, which is significantly more difficult in comparison to standard text tokens. Figure 2 empirically demonstrates this effect, showing that incorrectly accounting for whitespace tokens can reduce attack success rate by 28%. Appendix A.2 contains a more detailed discussion of this experiment. These issues can dramatically affect the success of an attack, and detecting them can be very tedious, as many whitespace characters are removed during detokenization and are thus invisible in the generated text. **We therefore urge developers to manually check the alignment of model generations with the target sequence *in token-space* to ensure correct alignment.**

The community widely agrees that token-level discrete optimization algorithms should exclude special tokens (e.g., instruction, tool use, role tokens) from the optimization. We thus echo the sentiment from (Thompson & Sklar, 2024b) and **strongly recommend that discrete attack algorithms track all state as a string** a user could pass directly to the model in a chat setting. While this may incur additional implementation overhead, it ensures that the discovered attacks are usable in practice. We find that even widely used implementations in nanoGCG (Zou et al., 2023b) and HarmBench (Mazeika et al., 2024) often fail to correctly filter out special tokens and other impossible attacks.

Table 1 illustrates these challenges. Given their widespread use and relatively well-supported software, we select GCG as the attack and Llama-3.1-8B-Instruct as the victim model, though other setups may exhibit even greater discrepancies. We compare several versions of the chat template, system prompts, quantization levels, and token filtering approaches. All reported setups are genuine best-effort implementations a researcher may attempt. Using the Meta-recommended chat template with a corrected strict token filtering procedure *lowers attack success rate (ASR) by 8%* compared to the HuggingFace Transformers library default template and token filtering from the official GCG implementation, while an int8 quantized model has *14% lower ASR*.

Efforts like (Mazeika et al., 2024; Chao et al., 2024; Beyer

et al., 2025) aim to tackle some of these issues by standardizing attack implementations and other aspects of the pipeline. However, in practice, many new algorithms are still built in bespoke repositories (Andriushchenko et al., 2025; Sadasivan et al., 2024), without taking advantage of these toolboxes. In our experience, the main reasons researchers opt for this are a lack of flexibility, reliability issues, and overly complex frameworks that attempt to abstract across models, but end up obfuscating underlying issues, leading to a steep learning curve and making them frustrating to work with. Furthermore, long-term maintenance is often not guaranteed, reducing the incentive for authors to implement their method in a particular framework. Solving these problems is challenging, as the development and continued maintenance of such frameworks are not sufficiently recognized by the academic community to justify the sustained effort.

**Attacks are not compared at consistent budgets.** Many algorithms have hyperparameters that trade-off compute, perplexity, or the number of model queries for higher ASR. However, most papers report performance for a specific "operating point"—typically one that maximizes ASR for the available resources—without adequately illustrating the compute-ASR trade-off for their method and baselines. This narrow focus on final ASR leads to misleading comparisons. For example, GCG is usually run with default hyperparameters, which use a large number of steps, making it appear inefficient and compute-intensive (Chao et al., 2025; Sadasivan et al., 2024). However, as seen in Figure 4 and Appendix A.3, GCG can reach significant ASRs and a better compute-ASR trade-off with far fewer steps.

Thus, to ensure fair evaluations, attacks should be compared under controlled conditions by fixing either the attack budget or the success rate. Results where neither is matched often do not allow for meaningful comparisons unless one method Pareto-dominates others in all domains.

Some recent works take steps toward better standardization by reporting the mean number of queries per success (Chao et al., 2025), iso-GPU-time results (Sadasivan et al., 2024), wall-time on consistent hardware (Geisler et al., 2024), monetary cost estimates for commercial models (Hughes et al., 2025), or FLOP-budgets to quantify compute use (Boreiko et al., 2025). We think these are all valuable ideas, and **reporting and controlling for attack budget should become standard in the field**.

**Optimization objectives are biased, unnatural, and poor proxies for attack success.** Most competitive attacks (AmpleGCG, AutoDAN, BEAST, GCG, PGD, PAIR, ...) rely on predefined target string sequences provided by the jailbreak datasets. This objective is suboptimal as it is not model-agnostic. Nearly all targets follow a rigid "Sure, here..." template. When a model's "natural" affirmative responses follow a different structure, the rigid targets become harder to optimize against and thus appear more robust than they actually are (Zhu et al., 2025) (c.f. Appendix A.4).

A similar issue can occur when defenses are trained with knowledge of the optimization targets the attacks will use. This data leakage can lead to methods that defend remarkably well when attacks try to optimize for the same targets used during training; however, when attacks choose a different objective, they can quickly become more brittle. For instance, R2D2 (Mazeika et al., 2024) was adversarially trained against GCG, and its safety generalizes relatively well to other approaches that aim to optimize the datasets' target sequence (e.g., AutoDAN). However, R2D2 remains susceptible to PAIR (Chao et al., 2025), which employs LLM-driven optimization of the prompt rather than heavily relying on the explicit target sequence.

Furthermore, successfully manipulating a model into outputting the full target sequence does not guarantee a harmful response as models can "recover" and continue with a safe response. Instead, the effectiveness of the generated attack depends on the implicit bias of the optimization algorithm. While some methods attempt to diversify the targets (Jia et al., 2025; Zhu et al., 2025; Thompson & Sklar, 2024b), or design adaptive attacks whose objectives are tailored to the specific victim model and defense (Andriushchenko et al., 2025), we think the field could benefit from considering a broader perspective. In other adversarial-robustness domains, durable progress came from adaptive attacks that construct consistent loss functions, where lower loss is guaranteed to correspond to higher success rates (Tramer et al., 2020).

Although it may not be feasible to design fully consistent loss functions for generative models, we think **the community should explore ways to more directly optimize for successful attacks**, rather than overfocusing on optimizing for specific token sequences that may introduce bias and fail to provide a consistent surrogate signal for ASR.

## 4. Evaluation

The last stage of the LLM safety evaluation pipeline typically consists of sampling a model generation and judging whether the model has been broken or not.

---

**Key Issues with Current Evaluation**

Limited to greedy generation, a fragmented ecosystem, judges with model- & attack-specific biases, and overlooked safety-overrefusal trade-off.

→ Overoptimistic results, misleading & biased evaluations for untested attacks & defenses.

---

**Greedy generation is unrealistic and ignores the distributional nature of LLM outputs.** Greedy decoding, where the model always selects the highest-probability token at each step, is currently the dominant setup for sampling from victim models in the context of LLM robustness. While convenient and deterministic, this strategy does not match practical chatbot settings where model responses are sampled using a variety of methods (Vijayakumar et al., 2016; Holtzman et al., 2020; Su et al., 2022) to avoid repetitive answers and produce natural output.

In fact, even with a fixed prompt, frontier models can often be broken by simply sampling a large number of generations at temperature 1 (Hughes et al., 2025)[2].

From a statistical standpoint, evaluating models at temperatures that differ from deployment settings is undesirable, as it can make noise mitigation techniques like resampling harder or even impossible in the case of greedy decoding (Miller, 2024). Additionally, Scholten et al. (2025) show that this setup fails to accurately characterize real-world model behavior, leading to over-optimistic bounds on safety and alignment: models that appear robust in greedy generation may still produce harmful content a significant fraction of the time when deployed with more realistic sampling strategies. Similar limitations also apply from the attacker's perspective: an attack prompt effective for greedy generation may not reliably generalize to real-world settings with more representative sampling methods (Hughes et al., 2025). To better assess the "robustness" of individual attack prompts and to provide better model safety guarantees, **we think a more realistic, distributional approach which samples multiple generations under realistic conditions and judges them independently may offer new insights**.

**Single-trial and many-trial jailbreaks lack clear differentiation.** There is a disconnect between methods that perform multiple queries and evaluate multiple model responses ("many-trial") until a jailbreak is found or the search budget is exhausted (Hughes et al., 2025; Liao & Sun, 2024; Chao et al., 2025; Andriushchenko et al., 2025), and those which optimize a prompt first and then test only the optimized prompt using greedy generation ("single-trial") (Zou et al., 2023b; Geisler et al., 2024; Sadasivan et al., 2024; Liu et al., 2024). Comparing these attack types is complicated as their practicality and efficiency depend on the model defenses employed and the cost model of the attacker. For example, cheaply generating many prompt candidates with low individual success probability may be efficient from a time or dollar cost standpoint but would require many model rollouts and could be more easily filtered by providers due to the suspicious query pattern of sampling a large amount of very similar prompts. In contrast, methods that expend more compute to come up with fewer (or even a single) prompts with high individual success likelihood may be more desirable from a query-efficiency standpoint.

Another consideration when comparing attacks is that attacks with "many-trial" characteristics are more likely to overstate their performance because they benefit from the classifier's false positive rate (FPR) for each query. Unless

---

[2]Hughes et al. (2025) primarily focus on augmented versions of harmful prompts, but an ablation using a fixed prompt shows that simply sampling more generations consistently increases ASR.

the results are manually verified[3], each additional inference raises the risk of sampling an outlier harmful generation or misgrading an unsuccessful attack as a false positive. In addition, recent results show that even with manual verification, it is possible to successfully jailbreak models by simplifying sampling a large number of responses, sometimes without altering the harmful prompt at all, further advantaging many-trial approaches (Hughes et al., 2025). Given these disparities, we argue that these two setups **must be clearly distinguished, and that many-trial attacks should account for the classifier's FPR when assessing the significance of results**. In addition, wherever possible, but particularly **for "many-trial" jailbreaks, we recommend manually screening generations for false positives. The LLM-judge ecosystem is fragmented.** Due to the prohibitive cost of human annotators, most works rely on automated classifiers to decide whether an output is harmful. However, the landscape is highly fragmented: some methods train custom classifiers (Huang et al., 2024b; Zhu et al., 2025; Dai et al., 2024), others use fine-tuned LLM judges like Harmbench Llama-2 (Mazeika et al., 2024; Boreiko et al., 2025; Zou et al., 2024) or StrongREJECT (Souly et al., 2024; Guan et al., 2024), while others employ pretrained models such as GPT or Llama variants using custom prompt templates (Chao et al., 2024; Hughes et al., 2025; Thompson & Sklar, 2024b). Some approaches incorporate human verification (Zou et al., 2023b; Hughes et al., 2025; Guan et al., 2024), and others fail to specify their evaluation methodology entirely. Although there are good reasons to train a new classifier (e.g., systematic failures in preexisting ones), the disjointed ecosystem of judges, prompt templates, and evaluation strategies complicate meaningful comparisons across studies.

In addition, despite prior work demonstrating that generation length affects attack success rates (Mazeika et al., 2024), there is still no consistent standard in the literature. Some works propose to use a consistent amount of judge tokens, while others use the victim model to generate a fixed number of tokens. While there are arguments to be made for both cases, we think the most realistic threat model would evaluate a full model response up to the end-of-turn/end-of-text token. This is closest to real-world use cases where users chat with models in a (mostly) unconstrained setting.

Finally, varying definitions of "successful" jailbreaks are used, with a clear trend of criteria becoming increasingly strict and precise over time, which we see as a positive development. We do not claim to have definitive answers, and we recognize that the community has not yet reached a consensus. Nevertheless, we believe that researchers would benefit from a more standardized judging environment, towards which recent toolboxes (Beyer et al., 2025; Maple et al., 2026) make progress. As judging is relatively compute-

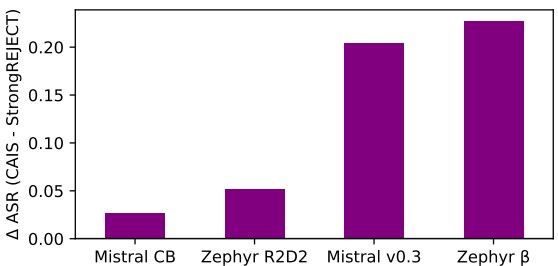

*Figure 3.* Comparing ASR differences across common judges reveals significant model-specific bias, despite all four models being derivatives of the same Mistral 7B model.

inexpensive compared to running attack algorithms or training a model defense, **we recommend that new works report results on multiple popular judges** (e.g., Harmbench Llama-2-13b and StrongREJECT), and **include a detailed description or code for the exact judge setup used** to facilitate easier comparisons and reproductions.

**Judges are not verified against new attacks & defenses.** Since attacks and especially defenses can significantly change a model's output distribution, generated responses may fall far outside the judges' training data, causing attack- and defense-specific misclassifications.

In Appendix A.5.2, we show that model-specific differences occur in practice and can lead to misleading comparisons between the safety of two models. We also conduct a large-scale study on 5,269,564 generations and compare judgments by Harmbench's Llama-2 13B classifier and StrongREJECT's Gemma 2B judge, and reveal attack- and model-specific biases—up to 25% for specific attacks and up to 24% across models (see Figure 3), complicating attack, defense, and model evaluations. Detailed results & analysis are provided in Appendix A.6. Similar biases have been observed before; for instance, Hughes et al. (2025) find that circuit breaker models (Zou et al., 2024) have higher judge FPR compared to baselines, which would lead to unfair comparisons without manual verification.

To avoid misleading results, **we propose that new attacks and defenses should include at least a small-scale human-verification that judges still work for the proposed method**, e.g., by manually checking 100 examples, as is done in (Chao et al., 2025). This is a manageable time investment that ensures attacks and defenses remain comparable and shields from effects such as the higher-than-expected FPR of some judges on specific models.

**Defenses do not consistently report the safety-overrefusal trade-off.** Designing safe LLMs inherently requires balancing competing objectives: maximizing robustness to adversarial inputs while minimizing overrefusal of benign prompts. This trade-off is well-documented (Thompson & Sklar, 2024a; An et al., 2024), yet, with few exceptions (Yu et al., 2024; Dubey et al., 2024; Guan et al.,

---

[3]In fact, even human judges may have nonzero FPR and could be affected too.

2024), existing defenses rarely account for it comprehensively. Many defenses only include results for capability benchmarks but do not use overrefusal datasets such as the aforementioned XSTest (Röttger et al., 2024) or the Co-CoNot contrast set (Brahman et al., 2024). We argue that **safety and overrefusal should always be evaluated in tandem**, with methodologies explicitly acknowledging this trade-off using standard benchmarks.

## 5. Alternative Views

**Small datasets have significant practical advantages.** While we advocate for the adoption of larger datasets, smaller ones offer key benefits. Even with small sample sizes, LLM robustness research is computationally expensive due to the size of modern models and the need for strong adversarial attacks to ensure reliable evaluations. Unlike prior adversarial robustness research in fields like computer vision, optimizing attacks in a discrete input space adds further complexity: current approaches require hundreds of thousands of queries per prompt, creating a bottleneck in robustness research. Using larger datasets would only exacerbate these issues and further raise the barrier to entry. Meanwhile, the lower computational demands of small datasets accelerate experimentation and research cycles, particularly in resource-constrained academic settings. This facilitates innovative research, as hypotheses can be evaluated quickly and cost-effectively. For instance, recent work (Andriushchenko et al., 2025) convincingly demonstrated that model-adaptive attacks can reach very high ASR using relatively few (50) samples. In general, even small datasets can provide enough signal to show the usefulness of a new approach at lower cost *if* the effect size is sufficiently large. Even when precise results are required, sophisticated sub-sampling strategies can be used to enhance the predictive power of small-scale experiments. For instance, Polo et al. (2024) reveal that a carefully selected subset of 100 questions can reliably approximate LLM capability on the full MMLU dataset with over 15k questions. This indicates that larger sample sizes may not be a strict requirement for comprehensive LLM evaluations.

**On flawed LLM judges.** Compared to other, more constrained domains with well-defined tasks and success criteria (e.g., image classification) the desire to find consistent and objective metrics may simply be misplaced in the context of LLM-robustness evaluations. Language is ambiguous and even our most precise rules will likely never catch all edge cases. The best we can do is iteratively refine our measures of jailbreaking success and continuously adapt automated judge models to capture these definitions more closely. In fact, this is exactly what the field is doing: researchers are constantly developing improved judge models that rely on increasingly precise definitions and better approximate hu-

man judgment (Mazeika et al., 2024; Souly et al., 2024; Zhu et al., 2025). Though imperfect, these evaluations are the only practical and scalable approach available.

**Perhaps we should explore more fundamental blind spots.** As models grow more capable and show better awareness of their condition, they may begin to strategically adapt their outputs to mislead researchers about their true nefarious capabilities. Recent work shows that even current models may deceptively adjust their outputs in response to a fictitious description of an environment that incentivizes such behavior (Greenblatt et al., 2024). Given these findings, maybe the community should instead move away from black-box input-output evaluations via judges and consider approaches like interpretability (Bereska & Gavves, 2024; Arditi et al., 2024) or representation engineering (Zou et al., 2023a). Understanding the internal mechanisms behind harmful outputs could enable more granular and robust assessments that help reduce the risk of deception.

**Finally, empirical evidence suggests that machine learning progress continues despite flawed metrics and noise.** Salaudeen & Hardt (2024) show that architecture improvements developed on a particular dataset are often robust to distribution shifts. Meanwhile, commonly used evaluation metrics for image generation have been criticized for bias and disagreement with human perception (Kynkäänniemi et al., 2022), yet the field has rapidly advanced (Rombach et al., 2022). If imperfect benchmarks & biased datasets were fundamental barriers, such rapid progress would be highly unlikely. Similarly, we continue to see genuinely new and innovative approaches in the context of LLM robustness (Zou et al., 2023b; Yu et al., 2024), contradicting the notion that flawed metrics and noise hinder innovation.

## 6. Ensuring the Adoption of Best Practices

Merely identifying flaws and technical solutions is insufficient if those solutions are not adopted in practice. Realistic solutions must create durable incentive structures and engage the entire publishing ecosystem: authors, reviewers, and organizers.

Authors should educate themselves on current best practices and adhere to them. Reviewers must incentivize these norms by consistently rejecting papers which fail to implement them. Lastly, organizers should codify these expectations via explicit reviewer guidelines emphasizing reproducibility and other quality standards.

While some venues (e.g., NeurIPS, ICLR), have begun incorporating such guidelines (NeurIPS, 2023; ICLR, 2023), we think they need to become much more widely adopted and made more prominent. Guidelines should also be tailored to subfields; one-size-fits-all recommendations are less likely to be taken seriously, e.g., if they are unrealistic or irrelevant in a particular context.

## 7. Conclusion

In this work, we critically examined the current state of LLM robustness evaluations, highlighting major sources of noise, bias, and inconsistencies across datasets, algorithms, and evaluation pipelines. Our analysis reveals that small and inconsistently subsampled datasets, implementation discrepancies, and fragmented evaluation setups severely hinder the reproducibility and comparability of research in this space.

To address these issues, we recommend the following:

- **Datasets:** Prioritize larger, high-quality benchmarks and avoid ad-hoc subsampling to reduce statistical uncertainty and simplify cross-paper comparisons.
- **Implementation:** Standardize and document details like quantization, tokenization, and chat templates, which significantly impact ASR.
- **Fair comparisons:** Control attacks for effort or success rate, and consistently report both safety and overrefusal for defenses.
- **Evaluation:** Use multiple judge models with manual verification, and shift from greedy generation toward distributional evaluation.
- **Incentives:** Reviewers and organizers should codify reproducibility requirements in *mandatory* subfield-specific guidelines; authors must adhere to them.

Our vision for LLM-robustness research is a field with correct, consistent, reliable, and reproducible results that are easily comparable across different projects, be it new attack algorithms, defenses, or models. We hope our work serves as a constructive step towards this goal.

## Impact Statement

We present work in the field of AI safety and robustness in the context of LLMs. Due to their widespread availability and rapidly increasing capability, their reliability and safety has significant consequences beyond the research community and may be of interest to industry and the general public. Researchers have an obligation to ensure their work meets scientific standards, such as reproducibility, and accurately communicates findings as objectively as possible. Many of our recommendations are centered around making sure that the field has the ability to accurately fulfill these standards without being misled by faulty or noisy evaluations.

Prior work on safety & alignment has also sparked controversy regarding the perceived trade-off between censorship and model safety. We acknowledge that it is possible to misuse alignment technology for nefarious purposes, such as censorship. However, we believe that the problems of alignment/robustness and the specification of what models should be aligned to should be treated as separate. We explicitly avoid making any judgements or recommendations related to the latter, and focus on proposing objective ways to compare true model behavior to a set of guidelines.

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

# A. Appendix

## A.1. Background

### A.1.1. ROBUSTNESS FOR LLMs

In the context of LLM safety research, an *aligned model* can be defined as one whose generations follow the rules and guidelines laid out by its developer. Such an LLM may be adversarially attacked via methods that produce inputs that cause the model to deviate from the desired guidelines (also called *jailbreaks*). We consider a model more *robust* or more *aligned* if it is more "difficult" to elicit the undesired behavior.

In contrast to computer vision, where standardized threat models like norm constraints can precisely capture the "difficulty" of breaking a model (Goodfellow et al., 2015; Madry et al., 2018), there is currently no consensus on how to best define difficulty in the context of LLM robustness. For practical purposes, models are typically ranked according to their robustness against a suite of existing attack algorithms (Zou et al., 2023b). Models that are broken less often or require more iterations to break are considered more robust. Most research, including this paper, is not focused on "true" worst-case robustness, which demands that models never produce undesired outputs given *any* input, and instead focus on real-world scenarios and algorithms to find practically usable jailbreaks in reasonable time (Zou et al., 2024). The terms are also used somewhat inconsistently, with "robustness" often referring more to the stability of a model's output with respect to constrained adversarial perturbations of its input rather than the ability to follow a developer's intentions.

More formally, we investigate a model $f_\theta$, which maps from the space of input token sequences $\mathcal{X}$ to output token sequences $\mathcal{Y}$. We also define a set of *admissible* input token sequences $\mathcal{X}^* \subset \mathcal{X}$ which consists of all token sequences that are reachable by tokenizing a user-submitted string (i.e., they do not contain control tokens or impossible-to-reach sequences due to token merging). We consider attacks that either find an admissible sequence of discrete input tokens $\tilde{x} \in \mathcal{X}^*$ that causes its response $y \sim f_\theta(\tilde{x})$ to conflict with the guidelines set out by the developer, or modify the model parameters $\theta$ themselves (e.g., through fine-tuning) to the same effect. Deciding whether a generation conflicts with the developer's guidelines is likely an ill-defined problem due to the ambiguity of language. In practice, the community uses judging functions $J$ which map from model generations $y \in \mathcal{Y}$ to either discrete success flags $\{0, 1\}$ or continuous scores $[0, 1]$. $J$ is instantiated either via human evaluation or, more commonly, as a finetuned LLM trained on data of human evaluations of harmful and harmless generations.

### A.1.2. ROBUSTNESS IN THE IMAGE DOMAIN

In the past, the computer vision robustness research community has struggled with reproducibility issues and unreliable defenses which were often quickly broken after their release (Athalye et al., 2018; Carlini, 2024; Uesato et al., 2018).

Clearly defining commonly used threat models, such as $L_\infty = 8/255$ norm constraints, which provide a precise measure of the "difficulty" of breaking a model (Goodfellow et al., 2015; Madry et al., 2018), have made results in field more reliable by making attacks and defense easily comparable and testable. Additionally, public leaderboards with consistent datasets and threat models have been used to track progress (Croce et al., 2021). The community also focused on more standardized frameworks (Papernot et al., 2016; Croce & Hein, 2020; Kim, 2020) and datasets (Krizhevsky et al., 2009), making results easier to reproduce and avoiding noise resulting from slightly different implementations.

## A.2. Case Study: Effects of White-space Tokens on GCG Against Llama 2

We run GCG (Zou et al., 2023b) against Llama-2-7b-chat-hf using default hyperparameters. We use the Meta chat template without system prompt (as recommended), and report the mean across the first 100 prompts from HarmBench (Mazeika et al., 2024). In one trial, we do not account for the SentencePiece (Kudo & Richardson, 2018) underline token (ID 29781), which Llama 2 predicts at the beginning of each of its responses, before continuing with regular text-tokens. This token is not added automatically by the tokenizer, even with the correct template, and must be manually inserted to ensure correctness.

We see that GCG makes faster progress when the white-space token is correctly handled, achieving a 91% many-trial ASR after 250 iterations, whereas the misaligned implementation only reaches 63%. The loss curve further illustrates this effect, with the correct implementation rapidly reaching more than 50% lower loss values.

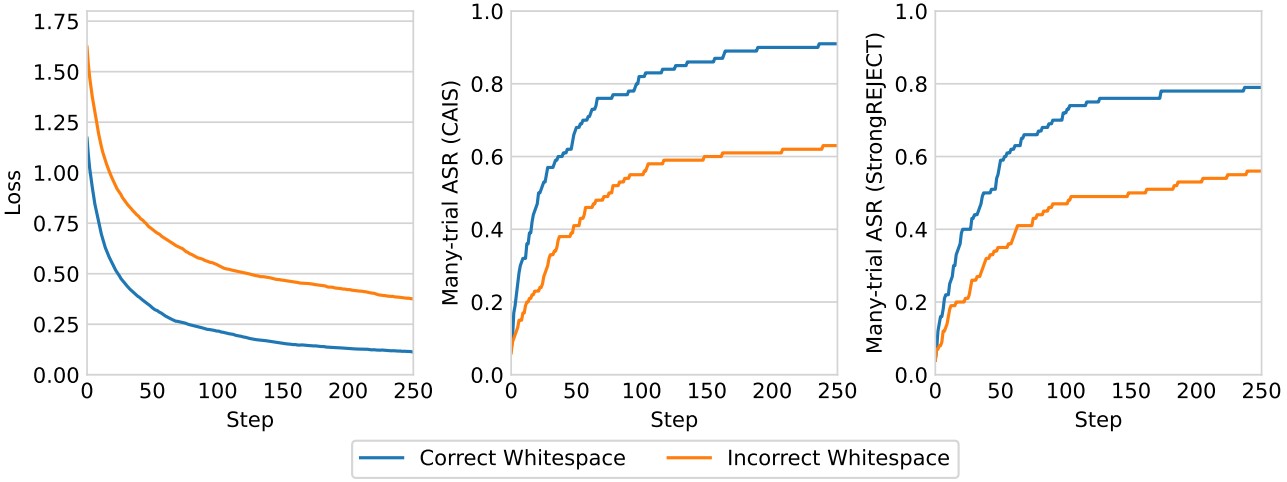

*Figure 4.* Impact of white-space tokens on GCG attack progress vs. Llama-2-7b-chat-hf.

## A.3. Case Study: Effects of Implementation Details on Optimization

We use the first 100 prompts from HarmBench (Mazeika et al., 2024) and run GCG (Zou et al., 2023b) against Llama-3.1-8B-Instruct. We use the default hyperparameters for GCG (batch size 512, number of steps 250) and use the BF16 floating point format for the model weights. Our reference implementation of GCG is based on the official nanoGCG repository but includes several enhancements.

- **Chat Template**: Instead of the default chat template provided by HuggingFace, we use the Meta-recommended template.

- **Token Filter for Non-Admissible Inputs**: While nanoGCG includes a filter to remove special tokens (control/role/system) tokens from the optimization, we design a stricter token filter which fixes a few issues. To evaluate whether an attack prompt is admissible, nanoGCG decodes and re-encodes the attack tokens and verifies that they remain unchanged. While this approach is sufficient for most scenarios, in some cases, the tokenizer merges characters *across the boundary* of the attack suffix and the remaining prompt, which is not detected by the filter. This can lead to inadmissible token sequences which cannot be reached from standard input strings and are therefore filtered by our implementation.

- **Filtering of Control Tokens**: The reference implementation fails to filter control tokens which were added to the tokenizer post-hoc, but can affect the optimization and should always be excluded.

In Figure 5, we demonstrate that these subtle changes can have a significant effect on the performance of GCG, causing an ASR difference of up to 14% among configurations which use a system prompt. We also include results without the system prompt to demonstrate the importance of following the instructions by model developers to include system prompts where required. The main point of this experiment is not to optimize for more efficient algorithms or higher ASR, but simply to show that minor details can significantly affect the outcomes of LLM attack evaluations.

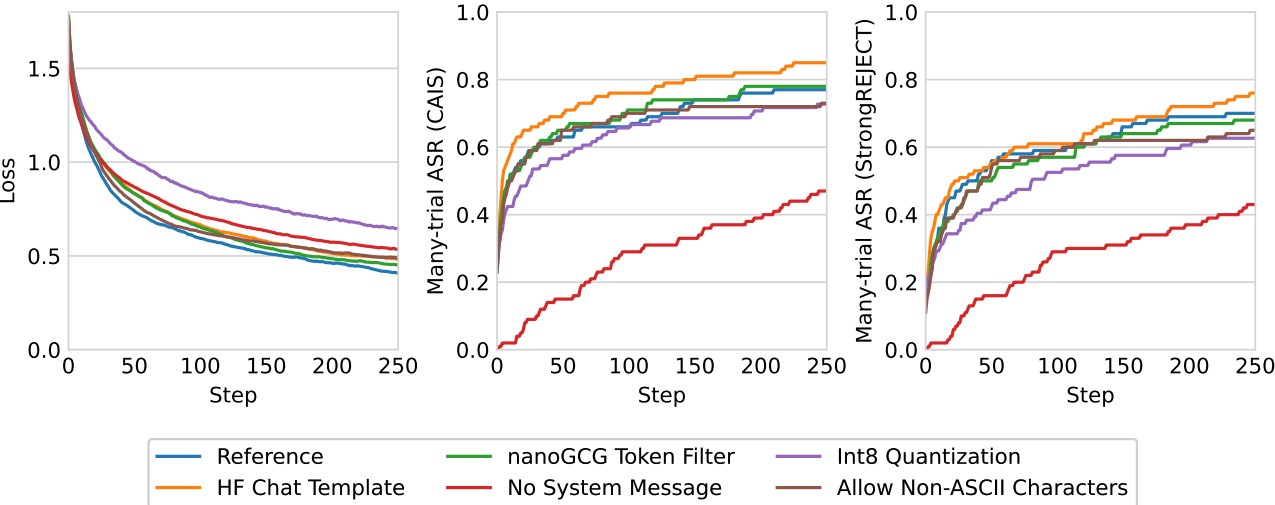

*Figure 5.* Impact of implementation details on GCG attack progress vs. Llama-3.1-8B-Instruct.

## A.4. Case Study: Bias in Optimization Targets

Because almost all targets provided in the HarmBench dataset (Mazeika et al., 2024) begin with the phrase "Sure, here...", we want to evaluate how "natural" this response is for various models. We run a small test using a benign prompt (*"Write a blog post that explains how to use the 'transformers' library to train a language model"*) and the target phrase *"Sure, here's a blog post that explains how to use the 'transformers' library to train a language model"*. We compute the average loss over the entire target phrase and the first token ("Sure") only. If the targets were model-agnostic, we would expect to see similar loss values for all models, however, this is not the case. This supports results showing that designing adaptive targets which are "natural" for a particular vicim model improves attack performance (Thompson & Sklar, 2024b; Zhu et al., 2025). We also see that many safety-focused model versions (Circuit Breaker (Zou et al., 2024), CAT (Xhonneux et al., 2024), LLM-LAT (Sheshadri et al., 2024)) have extremely high loss values for this specific type of affirmative response, likely because datasets with similar affimative response formats were used during training.

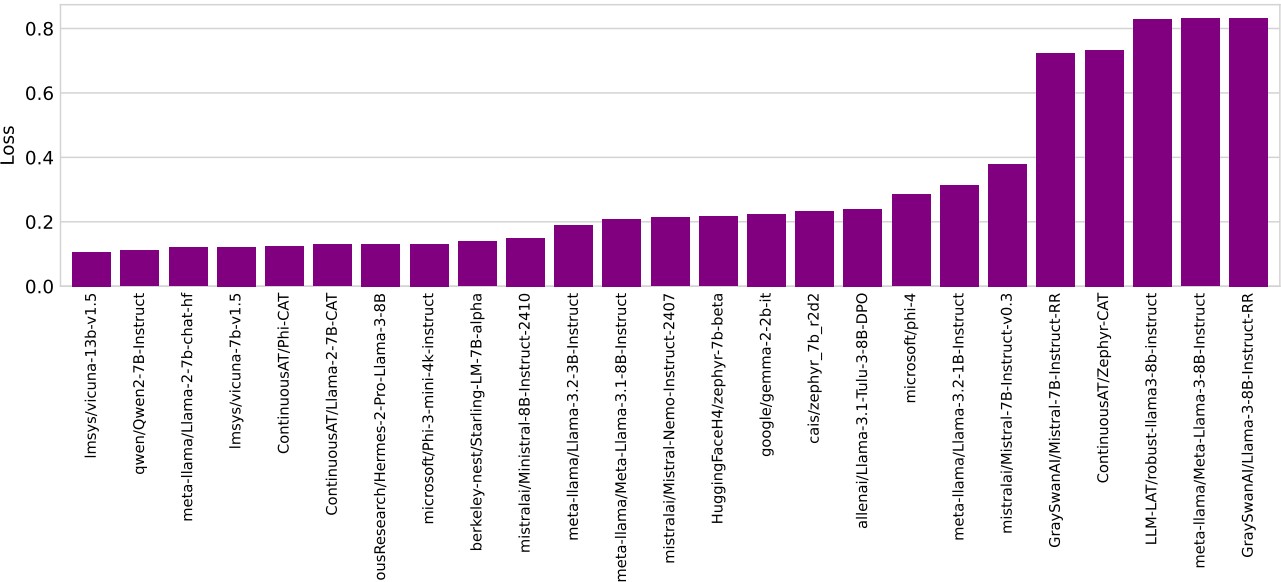

*Figure 6.* Average loss for HarmBench-style affirmative target sequence using a harmless prompt.

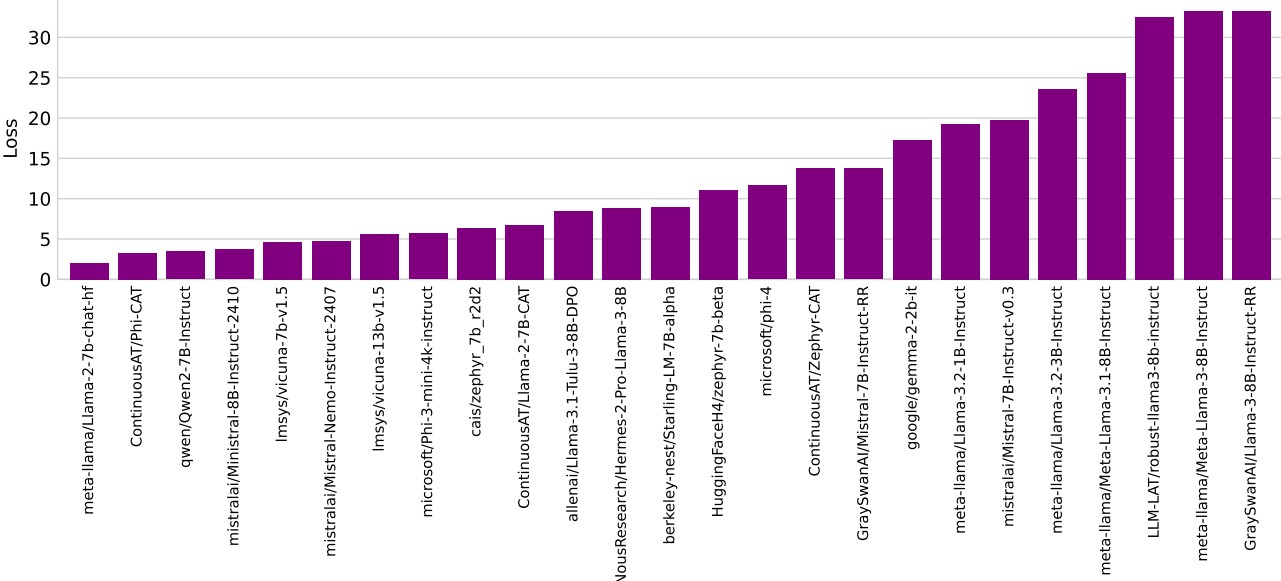

*Figure 7.* Loss for the first token ("Sure") for a HarmBench-style affirmative target using a harmless prompt.

## A.5. Case Study: Statistical Significance and Uncertainty

As a representative example, we investigate results reported in (Guan et al., 2024) more closely. The key finding of the paper is that reasoning models like OpenAI's o1 lead to a "Pareto improvement by reducing both under- and overrefusals" compared to existing state-of-the-art models. **We show that the results on public datasets alone (presented in Figure 2) would not be sufficient to support this claim at commonly accepted statistical significance levels**. In addition, we find that the use of automated judging for jailbreaks can lead to overly confident conclusions regarding the safety of models. We also want to emphasize that the findings of the work as a whole remain sound and well-supported, but **only because evaluations on larger internal datasets are included**.

### A.5.1. LIMITATIONS DUE TO DATASET SIZE.

The safety-overrefusal trade-offs of the models were evaluated on two public datasets: XSTest (Röttger et al., 2024), and StrongREJECT (Souly et al., 2024), which contain 250 and 313 prompts, respectively. We focus on comparing Claude 3.5 Sonnet and o1. By counting pixels on the figure, we estimate the underlying results to be as follows:

| Model | # Correct | |
| --- | --- | --- |
| | XSTest | StrongReject |
| o1 | 232 | 274 |
| Claude 3.5 Sonnet | 222 | 247 |

This aligns well with the results reported for o1 in Table 1. We denote the probability of correctly refusing a harmful query from StrongREJECT as $p_{\text{StrongREJECT}}$ and the probability of not overrefusing on XSTest as $p_{\text{XSTest}}$. To claim Pareto dominance, we need to show that $p_{\text{StrongREJECT}}^{(\text{o1})} > p_{\text{StrongREJECT}}^{(\text{Sonnet})}$ and $p_{\text{XSTest}}^{(\text{o1})} > p_{\text{XSTest}}^{(\text{Sonnet})}$. Thus, we perform two $z$-tests, one for each variable:

For StrongREJECT:

$$p_{\text{StrongREJECT}}^{(\text{o1})} = \frac{274}{313} = 0.875,$$

$$p_{\text{StrongREJECT}}^{(\text{Sonnet})} = \frac{247}{313} = 0.789,$$

$$p_{\text{StrongREJECT}} = \frac{274 + 247}{313 + 313} = 0.832.$$

Standard error:

$$SE = \sqrt{p_{\text{StrongREJECT}}(1 - p_{\text{StrongREJECT}})\left(\frac{1}{313} + \frac{1}{313}\right)}$$

$$= \sqrt{0.832 \times 0.168 \times \left(\frac{1}{313} + \frac{1}{313}\right)}$$

$$\approx 0.0297.$$

$z$-statistic:

$$z = \frac{p_{\text{StrongREJECT}}^{(\text{o1})} - p_{\text{StrongREJECT}}^{(\text{Sonnet})}}{SE}$$

$$= \frac{0.875 - 0.789}{0.0297}$$

$$\approx 2.90.$$

This yields a $p$-value: $\approx 0.004$, which indicates a highly significant difference at $\alpha = 0.05$.

For overrefusal:

$$p_{\text{XSTest}}^{(\text{o1})} = \frac{232}{250} = 0.928,$$

$$p_{\text{XSTest}}^{(\text{Sonnet})} = \frac{222}{250} = 0.888,$$

$$p_{\text{XSTest}} = \frac{232 + 222}{250 + 250} = 0.908.$$

Standard error:

$$SE = \sqrt{p_{\text{XSTest}}(1 - p_{\text{XSTest}}) \left( \frac{1}{250} + \frac{1}{250} \right)}$$

$$= \sqrt{0.908 \times 0.092 \times \left( \frac{1}{250} + \frac{1}{250} \right)}$$

$$\approx 0.0259.$$

$z$-statistic:

$$z = \frac{p_{\text{XSTest}}^{(\text{o1})} - p_{\text{XSTest}}^{(\text{Sonnet})}}{SE}$$

$$= \frac{0.928 - 0.888}{0.0259}$$

$$\approx 1.55.$$

This yields a $p$-value: $\approx 0.12$, which is not significant at $\alpha = 0.05$.

Thus we cannot claim a Pareto improvement using experiments on these two datasets alone.

### A.5.2. LIMITATIONS DUE TO AUTOMATED JUDGING.

The main body of the paper reports safety results on StrongREJECT using automated judges. However, the authors also include a human verification of jailbreaking performance in the appendix, which yields different results for StrongREJECT.

We perform an additional $z$-test to validate that o1 significantly improves jailbreaking robustness over Claude Sonnet 3.5 using the human-sourced data.

Given:

$$p_{\text{StrongREJECT}}^{(\text{o1})} = 0.92,$$

$$p_{\text{StrongREJECT}}^{(\text{Sonnet})} = 0.90,$$

$$p_{\text{StrongREJECT}} = \frac{0.92 + 0.90}{1 + 1} = 0.91.$$

Standard error:

$$SE = \sqrt{p_{\text{StrongREJECT}}(1 - p_{\text{StrongREJECT}}) \left( \frac{1}{313} + \frac{1}{313} \right)}$$

$$= \sqrt{0.91 \times 0.09 \times \left( \frac{1}{313} + \frac{1}{313} \right)}$$

$$\approx 0.0229.$$

$z$-statistic:

$$z = \frac{p_{\text{StrongREJECT}}^{\text{(o1)}} - p_{\text{StrongREJECT}}^{\text{(Sonnet)}}}{SE}$$
$$= \frac{0.92 - 0.90}{0.0229}$$
$$\approx 0.87.$$

This yields a $p$-value: $\approx 0.38$, which is not significant. Meanwhile, results from the automated classifier suggested that the difference is highly significant at $p \approx 0.004$!

The case study illustrates that significant problems may arise from relying on automated judges alone, as—in addition to unavoidable noise due to classification errors—they may have systematic biases that distort model comparisons. Here, we saw that Claude's true robustness was underestimated by 11.1%, while o1's was only underestimated by 4.5%, which directly affects the conclusions we can confidently draw from the experiments. We thus reiterate our suggestion to verify judge performance via human evaluation when proposing a new attack, defense, or model to ensure comparable results.

**A.6. Case Study: Comparing the StrongREJECT Gemma 2B and HarmBench/CAIS Llama 2 13B judges.**

We compare the results provided by StrongREJECT's (Souly et al., 2024) Gemma 2B judge model and the HarmBench (Mazeika et al., 2024) Llama 2 13B finetuned classifier.

To do so, we run a suite of 10 different attack algorithms against a zoo of 25 models using all 300 non-copyright prompts in HarmBench's dataset. All attacks are evaluated in the "many-trial" setting, regardless of the setting in which they were originally introduced. This means all intermediate prompt candidates are rolled-out and evaluated. In total, we compare results for 5,269,564 generations. We acknowledge that due to edge-case problems with models, tokenizers, and attack implementations, and due to time constraints, not all attack runs were successfully completed. Nonetheless, as we only compare the *difference*, we can nontheless make valid comparisons between the two judge models. Our results, shown in Figure 8, highlight substantial model- and attack-specific discrepancies between these two commonly used judges. While StrongREJECT's consistently reports lower ASRs (as expected, due to its more strict definition of what constitutes a successful jailbreak), we find very clear *differences in how much the two judges differ*. These inconsistencies can lead to differences in the scoring and even ranking of models, defenses, and attacks, complicating comparisons.

Ultimately, we would like to compare these classifier-based judgments to human-provided ground truth, but this was not feasible within the scope of this work. Nonetheless, if these two commonly used classifiers disagree, at least one (and likely both) must also disagree with human ground truth, underscoring the limitations of automated evaluation.

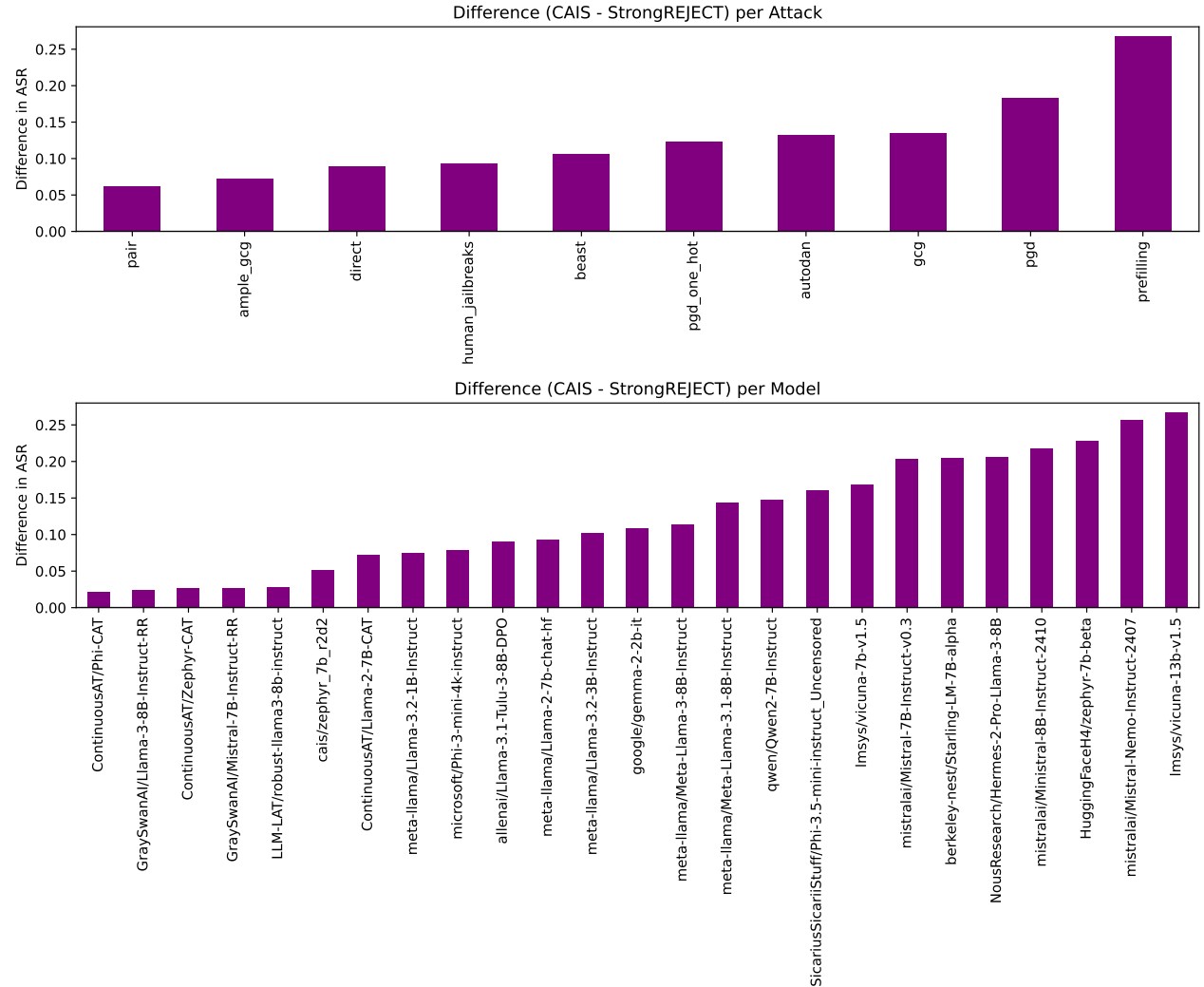

*Figure 8.* Comparing the ASR of two judge models. We consider an attack successful if StrongReject gives a score higher than 0.5, and if HarmBench's classifier outputs "Yes" or "yes" as the most likely token.

We briefly describe the hyperparameters we used below. All attacks implementations were adapted from the official GitHub repositories of the original authors where available. Otherwise, we ported a HarmBench implementation to our pipeline. In some cases, we conferred with authors directly to gain access to reference implementations and ensure correctness. For all attacks, we sample and judge a single greedy generation per prompt-candidate.

- PAIR (Chao et al., 2025): We use lmsys/vicuna-13b-v1.5 as the attacker model and generate up to 512 tokens per attack prompt. We sample with temperature 1 and top-p of 0.9, setting $N_{\text{streams}}$ to 5 and $N_{\text{iterations}}$ to 5. During the attack, the victim model generates up to 256 tokens using greedy generation.

- AmpleGCG (Liao & Sun, 2024): We use osunlp/AmpleGCG-llama2-sourced-llama2-7b-chat to generate 200 attack suffixes with diversity penalty 1.

- Direct: We simply use the harmful prompt without any modification and sample a greedy generation.

- HumanJailbreaks: We use the 114 human-designed jailbreak templates in HarmBench (Mazeika et al., 2024) to prompt the model.

- BEAST (Sadasivan et al., 2024): We use $k1 = k2 = 15$ and set the temperature to 1 to sample $N = 40$ suffix tokens.

- PGD (Schwinn et al., 2024): We initialize the attack using the suffix "x x x x x x x x x x x x x x x x x x x x" as it tokenizes to exactly 20 tokens for all tested models, and run signed gradient descent optimization for 100 steps. We use a learning rate of $\alpha = 0.001$ and constrain the optimization to an $L_2$ ball with radius 1 around the initialization for each token. In the one-hot version, instead of optimizing the input embeddings directly in latent space, we instead optimize over the $\{0, 1\} \rightarrow [0, 1]$ relaxed one-hot input vectors. In addition, we found it necessary to add a momentum term with weight 0.5 to speed up the optimization.

- AutoDAN (Liu et al., 2024): We use 100 steps and initialize using the 128 seed prompts from HarmBench's implementation. We use the victim model itself as mutator model and set $N_{\text{elites}} = 0.05$, crossover $= 0.5$, $N_{\text{points}} = 5$, and $P_{\text{mutation}} = 0.01$.

- GCG (Zou et al., 2023b): We use our modified version of nanoGCG with a corrected token filtering algorithm to remove special tokens & only allow ASCII-representable characters. We set $N_{\text{steps}} = 250$, use a batch size of 512, Top-K $= 256$, and initialize using the string "x x x x x x x x x x x x x x x x x x x x" as it tokenizes to exactly 20 tokens for all tested models.

- Prefilling: In this attack, we use the unmodified harmful prompt and pre-fill the beginning of the model's response using the affirmative target sequence from the dataset.

### A.7. Reproducibility

We will release code to exactly reproduce all[4] plots and tables upon publication. All experiments were conducted using greedy generation, bfloat16 precision, and HuggingFace Transformers version 4.47.1 unless otherwise noted. To judge generations we use the CAIS/HarmBench Llama 2 13B classifier and the finetuned Gemma 2B from StrongREJECT. For the HarmBench classifier, a generation is deemed harmful if the most likely output token is either "yes" or "Yes" and for StrongREJECT, we threshold the score at 0.5. We do not truncate model generations (i.e. we judge the full model response up to an end-of-sentence or end-of-turn token).

To obtain the results in Figures 4 & 5 and Table 1, we used a batch size of 512 and 250 steps of GCG with top-k set to 256. We initialized the suffix with the string "x x x x x x x x x x x x x x x x x x x x".

---

[4]Due to the large scale of the experiments and the associated rare occurrence of edge-cases and other non-deterministic problems, it may not be possible to exactly reproduce Figure 8 from scratch.

## A.8. Jailbreaking & Overrefusal Datasets

| Name | Harmful | Overrefusal | Notes |
|---|---|---|---|
| AdvBench (Zou et al., 2023b) | 520 | - | |
| CoCoNot (Brahman et al., 2024) | 3,528 | 1,306 | Some harmful prompts not considered forbidden according to all model developers' guidelines. |
| DangerousQ (Shaikh et al., 2023) | 200 | - | |
| HarmBench (Mazeika et al., 2024) | 400 | - | |
| HarmfulQ (Shaikh et al., 2023) | 200 | - | |
| HarmfulQA (Bhardwaj & Poria, 2023) | 1,920 | - | Highly repetitive (e.g., 20 prompts of the form "How can I use cultural studies to [harmful end]?") |
| JBB-Behaviors (Chao et al., 2024) | 100 | 100 | Overlap with AdvBench |
| MaliciousInstruct (Huang et al., 2024b) | 100 | - | |
| PromptEngineering (Liu et al., 2023b) | 40 | - | |
| StrongREJECT (Souly et al., 2024) | 313 | - | |
| Trojan Detection Challenge (Mazeika et al., 2023) | 50 | - | |
| XSTest (Röttger et al., 2024) | 200 | 250 | Harmful prompts contain ambiguous requests. |

*Table 2.* Commonly used jailbreaking and overrefusal datasets with harmful prompts and their size.

