# OpenReview forum: "Position: LLM-Safety Evaluations Lack Robustness"
_ICML.cc/2026/Position_Paper_Track — ICML 2026 Position Paper Track regular_

### Official Review · Reviewer_JhwK · 2026-03-11

**Significance:** 3
**Argument Clarity:** 4
**Rating:** 5
**Confidence:** 3

**Questions:**

1. The paper advocates expanding dataset scale and employing multiple sampling distributions for evaluation (as an alternative to greedy decoding) to mitigate noise and uncertainty. However, as noted within the text, given the pervasive computational resource constraints within academia, the threshold for rigorously implementing these recommendations is prohibitively high. Is it necessary for the community to establish a tiered assessment framework for security benchmarks? For instance, could a single benchmark offer both a “Flash version” (lightweight, featuring carefully subsampled small datasets, potentially employing greedy decoding, for rapid initial iterations and hypothesis validation) and a “Full version” (complete dataset + multiple samples from the distribution, serving as the rigorous benchmark for final paper publication)? What are the authors' views on this practical approach to balancing rigour with resource availability?

**Alternative Views Section:**

Yes

**Compliance With Llm Reviewing Policy A Conservative:**

Affirmed.

**Discussion Potential:**

3

**Final Justification:**

The authors have provided clarifications addressing my concerns. Based on these revisions, I maintain my original score.

**Paper Summary:**

This paper highlights significant reliability issues within the current landscape of large language model (LLM) safety alignment evaluation. These problems stem from multiple intertwined sources of noise, including small dataset sizes, inconsistent methodologies, and unreliable evaluation settings. This often prevents fair assessment and comparison of different attack and defence approaches, thereby hindering research progress. Regarding datasets, concerns include small sample sizes, inconsistent subsampling, lack of multi-round/multilingual testing, and ambiguous definitions of harmfulness. Concerning algorithms, implementation details (such as whitespace handling), uncontrolled attack budgets, and biased optimisation objectives; concerning evaluation, it criticises the impracticality of greedy decoding, the fragmented ecosystem of automated judges, and judges' biases towards specific attacks/models. Finally, the paper not only offers a balanced ‘alternative perspective’ section but also formulates a set of practical guidelines for future evaluations to reduce noise and bias.

**Position:**

Yes

**Position In Title:**

Yes

**Related Work:**

3

**Strengths And Weaknesses:**

** Strengths**

- Strong Empirical Evidence: The authors not only present critiques but also substantiate their arguments through multiple specific case studies. For instance, the experiments demonstrating that incorrect handling of SentencePiece whitespace characters reduces GCG attack success rates by 28% (Figures 2 and 4), alongside tests showing implementation details significantly impact Llama-3.1 attack outcomes (Figure 5), present highly persuasive data.

-  LLM safety represents a rapidly evolving yet poorly standardised domain. Highlighting the fragility of current benchmarking and evaluation metrics—such as the systemic bias in automated evaluators leading to misjudgements of certain models (Appendix A.5.2)—is both timely and critical.





**Weaknesses**

- The authors strongly recommend employing larger datasets and adopting distributional evaluation with multiple samples as an alternative to greedy decoding. While statistically sound, this approach entails an exponential increase in computational cost, rendering it particularly unfriendly to the resource-constrained academic environment. Although the paper acknowledges the advantages of small datasets in its ‘alternative perspectives’ section, it fails to propose a concrete solution to reconcile this contradiction.

**Support:**

4

---

> ### Author Rebuttal · Authors · 2026-03-30
>
> Thank you for the thorough review. We’re glad you found our arguments backed by "persuasive data" and would like to address the remaining concern around the tradeoffs between dataset size and compute demands.
>
> **W1/Q1: Tiered evaluation & compute demands of bigger datasets**\
> We took care to suggest many low-cost practical solutions (e.g., consistent subsampling) and explain our reasoning for why we believe more robust and standardized implementations and datasets will *reduce* compute and implementation costs in the long run as they would offer reliable and comparable reference results for existing work, reducing the need to reproduce prior work.
>
> That said, we think a tiered system could provide a reasonable tradeoff between rigor and computational requirements in the immediate term and describe this approach in L.117-119 (called 'dev' instead of 'Flash').
> Under such a framework, Flash versions could serve as legitimate tools for ablation studies and early-stage testing, while results on Full versions remain the expected standard for key claims.\
> The core risk of this approach is further fragmentation of the ecosystem: if results on Flash become accepted as sufficient, there will be a lack of incentives to evaluate on Full splits which fails to address the issue of noisy evaluations. To mitigate this, we think the community needs to adopt and uphold clear standards. While we acknowledge that norm-setting alone cannot guarantee adoption of Full evaluations, we believe that clearly defining what each tier can and cannot support (e.g., by embedding that distinction into benchmark tooling and documentation) lowers the barrier to good practice.
> Ultimately, enforcement rests with review processes and community expectations, which we think this framework helps clarify.
> In response to your comment, we have expanded the discussion of this tiered approach in the revised paper.

---

> > ### Author Rebuttal · Reviewer_JhwK · 2026-04-03
> >
> > The authors have provided clarifications addressing my concerns. Based on these revisions, I maintain my original score.

---

### Official Review · Reviewer_T2N2 · 2026-03-12

**Significance:** 3
**Argument Clarity:** 3
**Rating:** 4
**Confidence:** 3

**Questions:**

1. Can you prove that the differences in ASR results caused by variations in Implementation details are widespread?
2. The issues you raised are indeed important, but some are limited by the researchers' actual circumstances (such as computational resources and data constraints), while others stem from the current state of the LLM Safety research community (lack of larger and better datasets, overall consensus, etc.). To what extent do you think this paper will improve this situation?

**Alternative Views Section:**

Yes

**Compliance With Llm Reviewing Policy A Conservative:**

Affirmed.

**Discussion Potential:**

3

**Final Justification:**

The authors have adequately addressed all my key concerns in their rebuttal.

**Paper Summary:**

The paper argues that current LLM safety evaluation practices are unreliable due to noise and inconsistencies across datasets, algorithms, and evaluation procedures. It shows that commonly used safety datasets are small, fragmented, and often subsampled, producing large statistical uncertainty. Implementation differences, such as tokenization, chat templates, and quantization, can significantly alter attack success rates. Evaluations are further biased by inconsistent attack budgets, reliance on greedy decoding, and fragmented LLM judge systems with model-specific biases.

**Position:**

Yes

**Position In Title:**

Yes

**Related Work:**

3

**Strengths And Weaknesses:**

Strengths
1. The paper states a precise claim that current LLM safety evaluations lack robustness, and supports it with a structured pipeline analysis (datasets, algorithms, evaluation), making the argument easy to follow.

2. It focuses on concrete points the community cares about (jailbreak definitions, subsampling, implementation details, judge fragmentation), aligning with the practical research in this field.

3. The paper proposes practical remedies (report uncertainty, publish fixed subsamples, standardize attack budgets, use multi‑judge/manual verification, expand to multi‑turn and multilingual benchmarks) that can guide future work and conference policies.

Weaknesses:
1. Operational details for some recommendations are limited. Issues like “the strong focus on English examples” and “The LLM-judge ecosystem is fragmented” are identified correctly, but the paper lacks concrete, phased, low‑cost strategies that would help practitioners adopt the suggestions.

2. Some experiments have been tested on only one model (e.g., the whitespace characters in the Implementation details are only tested on Llama-2-7B-chat-hf), which only indicates the existence of the problem, but does not indicate whether it is widespread.

**Support:**

2

---

> ### Author Rebuttal · Authors · 2026-03-30
>
> Thank you for the thorough review, and for recognizing our "practical" recommendations and its potential to guide future work and policies.
>
> **W1: Limited operational details for some recommendations**\
> While we believe that we provide many actionable steps for most issues throughout the pipeline, we have added more recommendations [1, 2] for the areas you pointed out. For example, these projects include toolboxes with user-friendly and standardized implementations of judge models. In addition, we have strengthened our proposals for dataset construction with practical suggestions for reducing data leakage and enabling low-cost experimentation through predefined development splits (cf. our response to reviewer JhwK).
>
> **W2/Q1: Pervasiveness of issues**\
> We selected the whitespace issue as an illustrative example of an implementation detail that is particularly egregious for Llama 2 7B, but show similar problems for up to 25 different models (e.g., Figure 5 for implementation details and Figure 6 for judging inconsistencies), demonstrating that such issues are affecting a wide range of commonly used models.
>
> **Q2: Impact of our paper**\
> We think the community is lacking a set of accepted and consistent guidelines which contributes to a particularly noisy peer-review process. We envision a paper like ours to serve as a reference for practitioners and reviewers alike. Given that the existence of such a reference has been effective in other ML robustness domains [3], we are hopeful that our paper can have a similar impact for the LLM-safety community.
>
> [1] Maple, C.,et al. "A robust, defensible, and reproducible methodology for benchmarking single-turn jailbreak attacks on large language models." MLCommons. https://mlcommons.org/wp-content/uploads/2026/02/MLCommons___Security___Jailbreak_0_7_Paper_Collaborative.pdf (2026).
> [2] Beyer, Tim, et al. "AdversariaLLM: A Unified and Modular Toolbox for LLM Robustness Research." arXiv preprint arXiv:2511.04316 (2025).
> [3] Carlini, Nicholas, et al. "On evaluating adversarial robustness." arXiv preprint arXiv:1902.06705 (2019).

---

> > ### Author Rebuttal · Reviewer_T2N2 · 2026-04-03
> >
> > Thank you for providing the clarifications. The authors have adequately addressed all my key concerns in their rebuttal.

---

### Official Review · Reviewer_1M9y · 2026-03-13

**Significance:** 4
**Argument Clarity:** 4
**Rating:** 5
**Confidence:** 4

**Questions:**

1. What is the boundary between algorithms and evaluation?

2. What is the threat model?

3. Does data contamination impact the robustness? if so, it is necessary to discuss them.

**Alternative Views Section:**

Yes

**Compliance With Llm Reviewing Policy A Conservative:**

Affirmed.

**Discussion Potential:**

3

**Final Justification:**

The authors response addressed my concerns.

**Paper Summary:**

## Paper Summary

This position paper studies the robustness of current evaluation practices for LLM safety research. The authors argue that the evaluation pipeline for LLM safety—covering datasets,  algorithms, and evaluation—contains numerous sources of noise and bias that hinder fair comparisons across research works. The paper systematically reviews these issues, including small and fragmented datasets, inconsistent attack implementations, and unreliable evaluation setups such as biased LLM judges and unrealistic decoding strategies. It also proposes recommendations aimed at improving reproducibility for future LLM safety evaluations.

**Position:**

Yes

**Position In Title:**

Yes

**Related Work:**

2

**Strengths And Weaknesses:**

## Strengths

1. Timely and Relevant Topic.


LLM safety evaluations is an important issue for the research community.


2. Well Structured Paper.

This paper presents a clear decomposition of the LLM safety evaluation into three stages: datasets, algorithms, and evaluation. The structure is well organized, however, the boundary between algorithms and evaluation is not very clear.


3. Practical Recommendations.

The authors provide actionable suggestions to provide a fair  LLM safety evaluation.


## Weaknesses

1. Lack of Discussion on Data Contamination.

The major limitation of this paper is that this paper did not discuss the poential bias raised by data contamination, which is widely discussed in LLM evaluation [1, 2, 3, 4], and I think this issue also applies to safety evaluation.
Given that the work focuses on evaluation reliability, discussing possible contamination risks and mitigation strategies (e.g., dynamic benchmarks [5, 6], held-out datasets [7, 7]) would strengthen the paper.


2. Unclear Boundary Between Algorithmic Issues and Evaluation Problems.

The boundary between algorithmic issues and evaluation issues is not clearly justified. For example, the paper lists buggy toolboxes, ignoring the safety–overrefusal trade-off, inconsistent implementation details as algorithm issues. However, some of these issues appear more closely related to evaluation methodology rather than algorithm design.

3. Missing Threat Model and Lack of Adaptive Attack Analysis.

The paper discusses various issues in jailbreak attacks and robustness evaluation but does not clearly define a threat model. Without specifying the attacker and defender's capabilities and objectives, it becomes difficult to interpret the severity of the identified problems or the realism of proposed solutions.

In particular, the paper does not discuss adaptive attacks [9, 10], where attackers explicitly design strategies tailored to bypass a specific defense or evaluation framework.

4. Missing some related works.

[1] Benchmarking Large Language Models Under Data Contamination: A Survey from Static to Dynamic Evaluation


[2] Data contamination: From memorization to exploitation


[3] Investigating data contamination in modern benchmarks for large language models

[4] Leak, Cheat, Repeat: Data Contamination and Evaluation Malpractices in Closed-Source LLMs

[5] Dycodeeval: Dynamic benchmarking of reasoning capabilities in code large language models under data contamination

[6] Automating dataset updates towards reliable and timely evaluation of large language models

[7] Livebench: A challenging, contamination-limited llm benchmark

[8] Unseentimeqa: Time-sensitive question-answering beyond llms' memorization

[9] On Adaptive Attacks to Adversarial Example Defenses

[10] Jailbreaking Leading Safety-Aligned LLMs with Simple Adaptive Attacks

**Support:**

4

---

> ### Author Rebuttal · Authors · 2026-03-30
>
> We appreciate the reviewer’s detailed comments, and are happy they found our paper “timely” and our recommendations “practical”. We would now like to answer the reviewer’s questions.
>
> **W1/Q3: Data contamination**\
> We appreciate this important point and indirectly mention data contamination throughout the paper, for example in our discussion of overly rigid datasets and biased optimization targets, where we describe that prompts follow highly repetitive structures and that the vast majority of affirmative suffixes starts with an identical phrase like "Sure, here's ...". In Figures 6 & 7, we show that some safety-trained models are overoptimized against these particular targets and, because the same target patterns are used for evaluation, appear more robust than they actually are.
>
> In response to your comments, we revised these sections to **make the connection to data leakage much clearer** and have **added relevant references to provide context**. Lastly, we added a subsection on constructing datasets with lower risks of leakage and reorganized the "dataset subsampling" and "rigid dataset" sections. In the revised subsections we recommend providing predefined splits that can be used separately for training and evaluation, and propose releasing held-out, dated subsets in regular intervals that can be used to guarantee leakproofness in the future.
> > **Current datasets risk data leakage:** Overly rigid prompt templates, repetitive optimization targets, and inconsistent subsampling can increase the risk of leakage and artificially inflate measured robustness [1,2,3,4]. For example, as shown in Section A.4, an adversarially trained model using prompts and targets from a particular dataset and evaluated on the same data or from a similar distribution will appear more robust because patterns from the evaluation data distribution leak into the training set. To mitigate this, we recommend the creation of predefined, disjoint splits for robustness training and evaluation. In addition, we propose that future datasets should periodically release held-out evaluation sets in an ongoing fashion (e.g., once every quarter) to prevent data leakage [7,8].
>
> **W2/Q1: Boundary Between Algorithms and Evaluation**\
> We agree that the boundary between algorithms and evaluation was not stated clearly enough in the original version. In our framing, algorithms refer to the attacks and defenses themselves, i.e., components that change the behavior of the system under test, while evaluation refers to how robustness is measured and reported. The algorithm section discusses practical issues that arise when running attacks and defenses because these implementation choices ultimately affect the reliability of the evaluation. That said, we agree that the safety–overrefusal discussion concerns measurement completeness and reporting and thus moved it to the evaluation section.
>
> **W3/Q2: Threat model**\
> We have clarified our threat model in the revised version:
> > **Threat model.** We consider a standard, broadly defined threat model in which a safety-aligned language model $\pi_\theta$ is trained to refuse harmful requests. Given a set of harmful behaviors $\mathcal{B}$, an adversary seeks to elicit responses that substantively fulfill some $b \in \mathcal{B}$, either by manipulating the input prompt or by modifying the model parameters $\theta$ itself (e.g., through fine-tuning). Success is determined by a judge function $j(b, y) \in \{0, 1\}$ that evaluates whether a response $y$ fulfills the harmful target behavior.
>
> **W4: Adaptive attacks**\
> We agree that adaptive attacks are an important part of the threat landscape and should be discussed more explicitly. Our goal in this paper is not to evaluate one fixed attack class, but to identify failure modes in robustness evaluation that can distort conclusions across a wide range of attack and defense settings. These concerns remain equally relevant under adaptive attacks which we already cite in e.g., Section 5 and A.4 as they are related to issues like static and biased optimization targets.\
> In the revised paper, we have moved and refined the discussion of adaptive methods from the appendix into the "optimization objectives" section in the main body to increase visibility.

---

> > ### Author Rebuttal · Reviewer_1M9y · 2026-04-03
> >
> > The authors response addressed most of my concerns, if the author could provide more clear boundary definition between algorithm and evaluation layers, I will raise my score to 5.

---

> > > ### Author Response · Authors · 2026-04-06
> > >
> > > We appreciate the continued engagement and have revisited our paper's structure.
> > >
> > > Our framing adopts a split between decisions which influence the attack loop itself (i.e., the generation of attack artifacts like attack prompts) and those that affect the generation and assessment of model outputs.
> > > A clear intuitive test for our division is "can this problem be fixed entirely without re-running the attack algorithm, only using saved prompts from intermediate steps"? If the answer is 'no', we place the topic in the algorithmic section. For example, if we conduct an attack run at incorrect compute budget, we cannot increase the compute budget post-hoc to match a baseline which requires more compute - instead, we are required to re-run the attack with the correct budget. In contrast, we can easily sample generations using a different strategy by re-using the saved prompts or use a different judge model to evaluate generations without running attack code.
> > >
> > > While we can also see good arguments for resolving ambiguity in a different way, we currently view this as the best solution and have updated the section introductions and Figure 1 accordingly.

---

### Official Review · Reviewer_8UK8 · 2026-03-18

**Significance:** 3
**Argument Clarity:** 3
**Rating:** 4
**Confidence:** 3

**Questions:**

None.

**Alternative Views Section:**

Yes

**Compliance With Llm Reviewing Policy A Conservative:**

Affirmed.

**Discussion Potential:**

3

**Final Justification:**

Overall this is a valid position paper with clear problem structure and supporting evidence. The authors have addressed my concerns. I would maintain my positive evaluation of this work.

**Paper Summary:**

This paper provides extensive evaluation and discussion on existing research landscape of LLM safety evaluation. Through several case studies and results, this work identifies discrepancies in each of the evaluation pipeline, including dataset, algorithm and evaluation, and concludes with a set of recommended practices for ML community.

**Position:**

Yes

**Position In Title:**

Yes

**Related Work:**

3

**Strengths And Weaknesses:**

Strengths:
1. This paper zooms into specific technical details that significantly impact results, which provides clear and useful insights.
2. The arguments are well supported with evidences from multiple case studies.
3. The paper highlights actionable guidelines for each discussed pitfalls, which could serve as a valuable checklist for future researchers.

Weaknesses:
1. A significant portion of the paper focuses on dataset scale, subsampling, and lack of diversity. While these are valid points, they are always-correct criticisms in machine learning that apply to almost every domain. The discussion here feels somewhat unsurprising and lacks in-depth discussion of faced challenges that cause these issues, and how to address them.
2. The robustness discussed by the paper is mostly discrepancies in safety evaluation across works, which calls for a standardized evaluation infrastructure/practice. While many points are correct, they are not research-oriented but rather practice norms. It would be more significant to discuss on more fundamental challenges and gaps that make robust safety evaluation hard, beyond saying what people are doing differently.

**Support:**

3

---

> ### Author Rebuttal · Authors · 2026-03-30
>
> We thank the reviewer for the detailed feedback and are pleased they found our insights "clear and useful" and our proposed guidelines "actionable". We now hope to resolve the outstanding questions.
>
> **Dataset scale, subsampling, and quality**\
> We agree that dataset limitations are a common concern across ML. However, we believe they warrant special attention in the LLM-safety domain for two reasons.
> First, the extent of the problem is striking. We document specific cases where even widely cited safety papers rely on datasets that are less diverse and *much* smaller than would typically be considered statistically sound in other fields, in some cases as small as n = 40 (!). This weakens the signal-to-noise ratio to the point that it can hinder progress.
> Second, unlike more mature ML subfields, LLM-safety lacks established defaults which meet basic requirements (even without subsampling), partly due to an inconsistent publishing bar that reduces incentives to invest in rigorous evaluation infrastructure, and partly due to the absence of easy-to-use defaults (e.g., via open high-quality datasets with predefined splits). Additionally, beyond these important but relatively well-known concerns, our work includes significant novel contributions (e.g., regarding white-space issues, model-specific target biases, and judging discrepancies) and proposes concrete solutions for widespread issues that plague the field.
>
> **Research vs. practice norms**\
> We agree that advancing fundamental understanding of safety challenges is essential, and discuss this perspective in the 'Alternative Views' section. At the same time, even well-studied threat models such as text-based jailbreaks remain unsolved, suggesting that the bottleneck is not only a lack of novel research directions but also insufficient rigor in how existing ones are pursued. Improving practical norms is complementary to research on new paradigms, since methodological improvements like standardized protocols and representative datasets strengthen evaluation across threat models, including newly discovered ones.
>
> **Action**\
> In response to your review, we have revised the paper to more clearly motivate why dataset and evaluation challenges deserve focused attention in the safety domain specifically.

---

> > ### Author Rebuttal · Reviewer_8UK8 · 2026-04-06
> >
> > NA

---

### Decision · Program_Chairs · 2026-04-30

**Decision:**

Accept (regular)

**Comment:**

The position is well-motivated, the pitfalls of safety evaluation are correctly identified by the authors and the manuscript is well-written. Overall I am very positive about this paper. I would strongly suggest the authors to incorporate all the suggestions given by the referees.